# OptProver: Bridging Olympiad and Optimization through Continual Training in Formal Theorem Proving

Chenyi Li [* 1]   Yanchen Nie [* 1]   Zhenyu Ming [2]   Gong Zhang [2]   Kun Yuan [3]   Zaiwen Wen [4]

## Abstract

Recent advances in formal theorem proving have focused on Olympiad-level mathematics, leaving undergraduate domains largely unexplored. Optimization, fundamental to machine learning, operations research, and scientific computing, remains underserved by existing provers. Its reliance on domain-specific formalisms (convexity, optimality conditions, and algorithmic analysis) creates significant distribution shift, making naive domain transfer ineffective. We present Opt-Prover, a trained model that achieves robust transfer from Olympiad to undergraduate optimization. Starting from a strong Olympiad-level prover, our pipeline mitigates distribution shift through two key innovations. First, we employ large-scale optimization-focused data curation via expert iteration. Second, we introduce a specialized preference learning objective that integrates perplexity-weighted optimization with a mechanism to penalize valid but non-progressing proof steps. This not only addresses distribution shifts but also guides the search toward efficient trajectories. To enable rigorous evaluation, we construct a novel benchmark in Lean 4 focused on optimization. On this benchmark, OptProver achieves state-of-the-art Pass@1 and Pass@32 among comparably sized models while maintaining competitive performance on general theorem-proving tasks, demonstrating effective domain transfer without catastrophic forgetting.

## 1. Introduction

Large language models (LLMs) have demonstrated strong natural language mathematical reasoning on challenging competition problems, often producing complex multi-step solutions to Olympiad-level problems (Hendrycks et al., 2021; Lewkowycz et al., 2022; He et al., 2024). However, these natural language proofs lack machine-checkable guarantees, and subtle errors can be hard to detect. This motivates formal theorem proving, where statements and proofs are expressed in an interactive proof assistant and each step is verified by a trusted kernel, such as Lean (De Moura et al., 2015; Moura & Ullrich, 2021).

Building on this foundation, LLM-based formal theorem provers (Polu & Sutskever, 2020) have achieved impressive results on Olympiad benchmarks such as MiniF2F (Zheng et al., 2021). These theorem provers generally fall into two paradigms: whole-proof generation (Ren et al., 2025; Lin et al., 2025; Wang et al., 2025) and step-level proving (Lample et al., 2022). Whole-proof generators condition on the problem statement to output the entire proof script at once. While effective for simple statements, this approach often suffers from hallucination in long-horizon reasoning, where a single incorrect intermediate step invalidates the entire proof. In contrast, step-level proving treats proof construction as a sequential decision process (Yang et al., 2023). By iteratively proposing tactics and receiving immediate execution feedback, the model enables an interaction loop that facilitates tree search algorithms at inference time (Xin et al., 2025a;b; Wu et al., 2025b). This feedback-driven search paradigm facilitates domain transfer, as it allows the prover to extract and leverage local information directly from the proof assistant's responses.

Despite the impressive success of provers on general Olympiad mathematics, these models struggle significantly when applied to optimization. Optimization stands as a cornerstone of applied mathematics and scientific computing, underpinning advancements in machine learning and operations research. Current proof models struggle to formalize convexity properties or establish the algorithmic convergence guarantees central to optimization theory. This performance gap motivates the need for domain adaptation, transferring knowledge from Olympiad mathematical

---

[*]Equal contribution [1]School of Mathematical Sciences, Peking University [2]Huawei Technologies [3]Center for Machine Learning Research, Peking University [4]Beijing International Center for Mathematical Research, Peking University. Correspondence to: Kun Yuan <kunyuan@pku.edu.cn>, Zaiwen Wen <wenzw@pku.edu.cn>.

*Proceedings of the 43rd International Conference on Machine Learning*, Seoul, South Korea. PMLR 306, 2026. Copyright 2026 by the author(s).

corpus to a specific undergraduate domain without suffering from catastrophic forgetting (Kirkpatrick et al., 2017). While techniques such as data mixing and regularization have proven effective in natural language processing and informal reasoning (Lewkowycz et al., 2022; Ahn et al., 2024), they remain insufficient for the unique challenges of formal verification.

This insufficiency arises from a unique constraint in formal proving: the rigor of the verifier. Unlike natural language domain adaptation, where subtle reasoning errors are tolerable, formal proving demands syntactic correctness at every step. This binary feedback creates a severe learning barrier when adapting to the specialized definitions and syntax of optimization libraries. We highlight two major limitations of naive adaptation. First, the model suffers from a loss of general skills. As the model struggles to adapt to the specialized syntax of the new domain, it tends to drastically forget the general reasoning skills it learned previously. The adaptation to new definitions comes at the cost of its foundational capabilities. Second, the model lacks effective strategies. While the model may successfully adapt to the rules of the new domain with generating valid steps, it fails to adapt to the strategies required to solve problems. It frequently proposes tactics that are syntactically correct but strategically aimless in the new context, leading the proof search into dead ends rather than a valid solution.

To overcome these challenges, we present OptProver, a formal theorem proving model designed to bridge the gap between Olympiad-level problems and undergraduate optimization. We propose a robust pipeline that ensures stable continual training from foundational proving skills to specialized Lean 4 optimization libraries. OptProver combines optimization-focused data construction with a self-play curation mechanism. Besides, drawing inspiration from preference-based learning (Christiano et al., 2017; Ouyang et al., 2022), we introduce a verifier-driven, utility-aware preference optimization strategy, enhanced by perplexity-weighted direct preference optimization to stabilize model training. Our method explicitly penalizes correct-but-unhelpful tactics. To rigorously evaluate our approach, we construct a novel benchmark consisting of 400 problems based on Optlib (Li et al., 2025; 2026). On this benchmark, OptProver substantially improves Pass@1 and Pass@32, overall finishing over half of the problems, achieving state-of-the-art in-domain performance among comparably sized models while maintaining high performance on general theorem-proving benchmarks.

Our main contributions are listed as follows.

- We formulate continual training for step-level provers as a domain shift problem from Olympiad proofs to optimization problems. We provide strong empirical evidence to demonstrate that naive continual training

is susceptible to fragility and catastrophic forgetting.

- We develop a verifier-driven, utility-aware preference learning method to penalize correct-but-unhelpful tactics. This is further enhanced by perplexity-based optimization strategy. This approach, together with delicate data curation, improves search efficiency and strengthens in-domain adaptation without sacrificing general proving capabilities.

- We establish OptBench, a novel benchmark designed to evaluate models' capabilities in generating formal optimization proofs. Extensive experiments on OptBench show that our model achieves a success rate over 55%, demonstrating a substantial advantage over formal provers of comparable scale without compromising performance on original Olympiad problems.

## 2. Preliminaries and Related Works

**Step-level formal proving and Expert Iteration.** Step-level formal proving is typically cast as a tree-search process (Wu et al., 2025b; Shen et al., 2025) that interacts with a proof assistant (e.g., Lean 4). In this setting, a language model $\pi_\theta$ proposes tactics to transition between proof states, with search conducted under a fixed computational budget. To train such provers, expert iteration (EI) has become a widely adopted framework, which alternates between model-guided search to discover proofs and supervised updates to learn from them. Formally, let $\mathcal{D}_{\text{succ}}$ denote the collection of state-tactic pairs $(s, a)$ extracted from verified proofs found during the search. The policy is refined by minimizing the negative log-likelihood on these successful trajectories:

$$\mathcal{L}_{\text{SFT}}(\theta) = -\mathbb{E}_{(s,a)\sim\mathcal{D}_{\text{succ}}}\big[\log \pi_\theta(a \mid s)\big]. \quad (1)$$

Unlike static supervised learning, the dataset $\mathcal{D}_{\text{succ}}$ in this framework is constructed iteratively (Xin et al., 2025b). As the policy evolves, it generates new candidate proofs via search algorithms. Trajectories that pass the formal verifier are then aggregated into the training set, allowing the model to continuously learn from its own successful reasoning.

**Perplexity-based Data Selection.** Perplexity is widely used as a metric to filter pretraining data or align it with a target distribution (Moore & Lewis, 2010). Specifically, researchers often calculate the difference in perplexity between a target model and a general reference model to score and select relevant documents (Xie et al., 2023). In the context of instruction tuning, similar scoring methods are used to filter out low-quality or simple queries, proving that a small amount of curated data can match the performance of larger datasets (Chen et al., 2023). Moving beyond the document level, recent studies have shifted to the token level, suggesting that different tokens within a single text

contribute differently to learning (Lin et al., 2024). This token-level approach has been effectively applied in active learning (Azeemi et al., 2024) and standard fine-tuning, where learning from low-perplexity tokens helps reduce catastrophic forgetting (Wu et al., 2025a).

**Direct Preference Optimization.** Direct preference optimization (DPO) (Rafailov et al., 2023; Meng et al., 2024; Azar et al., 2023) provides a robust framework for optimizing policies directly from pairwise comparisons, serving as a stable alternative to standard reinforcement learning from human feedback by avoiding an explicit reward model. Rather than maximizing a scalar reward, DPO trains the policy $\pi_\theta$ to prefer chosen responses over rejected ones by increasing their relative log-ratio score. Concretely, optimization is performed relative to a frozen reference model $\pi_{\text{ref}}$, and the implicit preference signal is defined by the log-probability ratio between $\pi_\theta$ and $\pi_{\text{ref}}$. Specifically, for a given state $s$ and candidate action $y$, DPO defines a log-ratio score function:

$$s_\theta(s, y) = \beta \log \frac{\pi_\theta(y \mid s)}{\pi_{\text{ref}}(y \mid s)}, \tag{2}$$

where the coefficient $\beta > 0$ regulates the scale of the update and effectively controls the deviation from the reference distribution. Consequently, given a preference dataset consisting of state $s$ and pairs of preferred $(y_w)$ and dispreferred $(y_l)$ actions, the pairwise DPO loss is formulated as:

$$\ell_{\text{DPO}}(s, y_w, y_l) = -\log \sigma\Big(s_\theta(s, y_w) - s_\theta(s, y_l)\Big). \tag{3}$$

This objective increases the relative log-likelihood of the preferred action $y_w$ over $y_l$, while the anchoring to $\pi_{\text{ref}}$ implicitly acts as a regularizer to preserve the stability and quality of the generated outputs.

## 3. Interactive Data Distribution Shift

To build a proving model expert in both optimization and Olympiad, we utilize BFS-Prover-V2 (Xin et al., 2025b) as our initial model. A natural approach is to curate large-scale formal data from undergraduate basic to optimization, and continually train the model on the related data. However, we find that direct continual training on such curated corpora can degrade proof performance rather than improve it. We observe that the efficacy of neural theorem provers is fundamentally constrained by the alignment between the training distribution and the proof skills the model learned during pretraining.

A critical phenomenon emerges in this setting, which we term *interactive distribution shift*. While static evaluation metrics such as perplexity may suggest that a model has been successfully adapted to a new domain, they provide

limited guidance for step-level proving, where success is determined by interactive proof search. Crucially, each data source induces its own proof style, including tactic idioms, lemma-usage patterns, and rewriting conventions. Continual training on a new style shifts the model's tactic prior and branch ordering during search, increasing the probability of locally valid but unproductive actions and causing catastrophic forgetting of the higher-leverage heuristics needed for the original proof style.

### 3.1. Curating the Optimization Corpora

To study how corpus affects interactive proving, we curate a dataset spanning foundational undergraduate mathematics to specialized optimization theory.

**Data Sources.** We utilize two primary data streams related to undergraduate optimization. The data sources come from formalized literature textbooks. We generate formalized versions of standard textbooks, including Convex Analysis (Rockafellar, 1997) and Real Analysis (Tao, 2006) via auto-formalization using large language models. This results in roughly 2k theorems and 40k state-tactic pairs, which we denote as $\mathcal{D}_{\text{book}}$. We utilize state-tactic extraction to capture the proof state $s$, including all goals and local hypotheses, immediately prior to the application of a tactic $a$. This guarantees that every training pair $(s, a)$ represents a verifiable transition.

### 3.2. The Decoupling of Perplexity and Utility

A prevailing assumption in large language model is that minimizing perplexity on a target corpus correlates with downstream task performance. We challenge this assumption in the context of formal reasoning.

Formally, for an output solution $\mathbf{y} = (y_1, \ldots, y_T)$ generated conditioned on a statement $\mathbf{x}$, where $y_i$ denotes each token of the output $\mathbf{y}$, the perplexity is the geometric mean of the inverse conditional probabilities:

$$P_{\text{ans}}(\mathbf{y} \mid \mathbf{x}) = \exp\left(-\frac{1}{|\mathbf{y}|} \sum_{t=1}^{|\mathbf{y}|} \log p_\theta(y_t \mid \mathbf{x}, y_{<t})\right). \tag{4}$$

**The Low-Perplexity Trap.** As detailed in Table 1 and Figure 1, our corpora exhibit distinct statistical profiles relative to the model's priors. While the *Real Analysis* formalizations achieve a perplexity of 2.875, closely mirroring the 2.606 observed in high-quality expert iteration traces, the *Convex Analysis* dataset exhibits a notably higher average perplexity of 3.2. This statistical divergence signals a severe distribution shift in the optimization-focused material.

Contrary to the expectation that standard textbook formalizations would inherently serve as effective training data, Figure 2 illustrates that direct supervised learning on these

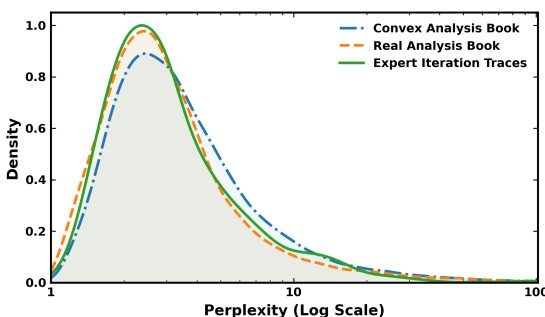

*Figure 1.* The perplexity distribution of BFS-Prover-V2-7B on different data corpora.

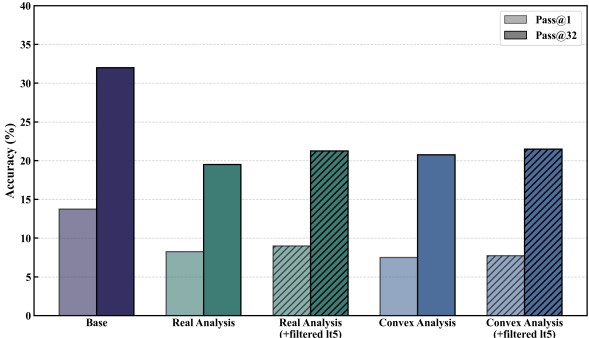

*Figure 2.* Performance degradation of OptBench under naive SFT. Base denotes the original BFS-Prover-V2-7B model. Despite filtering training data by perplexity, fine-tuning on whole-proof traces still diminishes the model's proving capability relative to the baseline, underscoring the limitations of direct supervised fine-tuning.

corpora leads to significant performance degradation on validation benchmarks. This decline is particularly pronounced in Pass@1 when incorporating the higher-perplexity *Convex Analysis* data, suggesting that the model struggles to assimilate the different reasoning style without compromising its baseline capabilities. We also apply a strict filter to retain only samples with a sequence perplexity lower than 5. However, while this filtration slightly mitigates the severity of catastrophic forgetting, the negative trend persists. This leads to a crucial observation. Training on external data, especially when it exhibits higher perplexity like *convex analysis*, induces negative distribution shifts that degrade reasoning capabilities, challenging the assumption that perplexity minimization is a sufficient proxy for downstream utility in formal provers.

**Analysis: Validity vs. Utility.** We attribute this failure to a fundamental decoupling between syntactic validity and proof utility, driven by a stylistic mismatch between the tactics in the new corpora and the model's prior training. While the model successfully minimizes perplexity by learning the surface form of the new data, the tactic idioms in new textbooks differ significantly from the Olympiad-style

*Table 1.* Perplexity across corpora (Avg. PPL: average perplexity per sequence).

| Source | State-tactic pairs | Avg. PPL |
|---|---|---|
| Convex Analysis Book | 21,021 | 3.201 |
| Real Analysis Book | 22,238 | 2.875 |
| Expert Iteration Traces | 121,322 | 2.606 |

proofs the base model was optimized for. Consequently, the model learns to generate tactics that are syntactically valid and follow the style of the new distribution, but often fail to advance the specific proof state effectively. In the tree search, these tactics act as distractors. They occupy high-probability branches in the search tree, exponentially increasing the branching factor and diluting compute resources without contributing to proof resolution.

## 4. Stabilization of Continual Training

In this section, we present the training methodology used to develop OptProver, a neural theorem prover specialized for optimization while maintaining robust ability on Olympiad-level problems. Starting from an Olympiad-purpose base model, we aim to instantiate OptProver by navigating the trade-off between acquiring new domain-specific syntax and preserving existing proof abilities. To achieve this, we propose a multi-stage continual training framework that addresses the challenges of distribution shift and catastrophic forgetting. This framework consists of three integrated components: large-scale self-distillation for valid trajectory discovery, and two complementary preference optimization strategies, utility-aware preference optimization and perplexity-weighted DPO, to enhance ability and search efficiency.

### 4.1. Large-Scale Self-Distillation via Expert Iteration

A primary risk in domain adaptation is that the model may fail to learn effectively from textbook proofs if the tactic style is too distinct from its pre-trained distribution. To mitigate this, we do not simply fine-tune on raw data. We treat the statements in Mathlib as the foundational seeds for large-scale exploration. Mathlib serves as a massive, unified library of formalized mathematics covering diverse domains from analysis to topology. For every theorem $T$ in this corpus, we execute a proof search using the current policy $\pi_{\theta_k}$. This process effectively translates the static mathematical knowledge of Mathlib into the model's own tactic language. By filtering for successfully closed proofs, we construct a dataset $\mathcal{D}_{\text{lib}}$ consisting exclusively of trajectories that are mathematically correct and discoverable by the model.

However, to equip the model with optimization domain knowledge, relying solely on Mathlib is insufficient. We

must integrate data from structured textbooks. Hence, the formalization of Convex Analysis and Real Analysis is incorporated into a supplementary dataset $\mathcal{D}_{\text{book}}$. Furthermore, to enable the model to master exercise solving, specifically focusing on applying techniques and theorems learned from textbooks, we add a problem-centric corpus. We translate nearly 7k natural language problems concentrated on analysis and linear algebra from DeepTheorem (Zhang et al., 2025) to formal language in Lean, constituting the problem corpus $\mathcal{D}_{\text{pro}}$. Formally, the policy is updated via negative log-likelihood minimization on the answers $\mathcal{A}$ derived from this mixture $\mathcal{D}_{\text{lib}} \cup \mathcal{D}_{\text{book}} \cup \mathcal{D}_{\text{pro}}$:

$$\mathcal{L}_{\text{SFT}}(\theta) = -\mathbb{E}_{(s,a)\sim\mathcal{A}} \left[ \log \pi_\theta(a \mid s) \right]. \qquad (5)$$

Subsequently, we extend this framework into a continuous loop of expert iteration. By deploying the updated policy derived from the previous stage, we repeatedly re-explore the theorem space to get new solutions. This iterative process allows the model to harvest fresh, higher-quality solution trajectories that were previously beyond its reach. These new data points are integrated back into the training pipeline to perform continuous parameter updates, ensuring that the model steadily refines its tactic generation capabilities while maintaining training stability.

## 4.2. Utility-Aware Preference Optimization

While self-distillation aligns the model with the valid tactic, standard likelihood maximization fails to explicitly penalize the distractor modes identified. In the context of tree search, valid but unhelpful tactics are particularly detrimental as they act as search sinks, consuming compute budget without resolving the goal. To address this, we introduce a preference learning mechanism that leverages the rigorous feedback signals provided by the Lean verifier to strictly order tactic utility.

**Utility and Preference Construction.** We formally define the utility scale $u(s,a) \in \{0, 1, 2\}$ for state-tactic pairs generated during search. Tactics are categorized based on their downstream search outcomes: proven transitions ($u = 2$) denote steps belonging to a verified closed proof path; invalid transitions ($u = 0$) correspond to compilation or runtime errors. Crucially, we identify an intermediate class of stagnant transitions ($u = 1$). These tactics are syntactically valid and successfully modify the proof state, yet lead to subtrees that exhaust the search budget without resolution. For instance, for vectors $u, v \in \mathbb{R}^n$, assuming $u_i > 0$ and $v_i > 0$, consider proving the following inequality

$$\sum_i \left( u_i \log \left( \frac{u_i}{v_i} \right) - u_i + v_i \right) \geq 0.$$

A typical stagnant tactic applies the linear algebra rule `Finset.sum_add_distrib` to split the expression into $\sum_i (u_i \log(u_i/v_i) - u_i) + \sum_i v_i$. While valid and locally attractive, this move creates a dead end. Although the model can easily prove the subgoal $\sum v_i \geq 0$, separating these terms breaks the structural integrity of the inequality. Without the $v_i$ terms, the remaining expression is no longer guaranteed to be non-negative, rendering the proof impossible to complete. Such tactics are highly detrimental. They may exhibit high likelihood under the pre-trained model, which is low perplexity but degrade inference efficiency by causing an exponential explosion of ineffective branches that lead to dead ends.

**Optimization Target Construction.** We construct the preference dataset $\mathcal{D}_p = \{(s, a_w, a_l)\}$ by strictly separating tactics generated within the search trees generated during self-play. For every state $s$ situated on a verified proof trajectory, we designate the tactic $a^*$ that advances towards the confirmed solution as the winner $a_w$. The negative candidates $a_l$ are sampled from the sibling branches, encompassing both tactics resulting in syntax errors and those that are locally valid yet globally failed to yield a proof within the search budget.

Crucially, this construction enforces a strict preference for realized efficiency over potential validity. While some sibling branches $a_l$ might theoretically be provable given infinite compute, their failure to resolve within the budget identifies them as computationally expensive or stagnant relative to the winner. By optimizing the relation $a_w \succ a_l$, we penalize these harder-to-traverse paths alongside invalid ones. This compels OptProver to ignore valid-but-convoluted distractions and concentrate probability mass on the most efficient trajectories that drive verifiable proof resolution.

## 4.3. Perplexity-Weighted Preference Optimization

The preference data often contains tokens that exhibit high perplexity under the reference model $\pi_{\text{ref}}$. These high-perplexity tokens typically correspond to either data noise or samples that are significantly out-of-distribution relative to the reference policy. Optimizing on these tokens where $\pi_{\text{ref}} \approx 0$ can lead to numerical instability and high-variance gradients, as the implicit reward term involves the ratio $\pi_\theta/\pi_{\text{ref}}$. To mitigate the optimization difficulties caused by these low-probability tokens in preference learning, we propose perplexity-weighted DPO (PW-DPO), which modulates the loss contribution based on token-level alignment with the reference distribution.

**Token-Level Weighting Mechanism.** We quantify the deviation of each token from the reference policy using token-level perplexity. For a token $y_t$ at decoding step $t$, the perplexity is defined as:

$$\text{PPL}_t = \exp \left( -\log \pi_{\text{ref}}(y_t \mid x, y_{<t}) \right). \qquad (6)$$

Tokens with high $\text{PPL}_t$ indicate regions where the reference model has low probability mass. To prevent these tokens from dominating the gradient updates, we assign a scalar weight $w_t$ using a bounded inverse-perplexity function:

$$w_t = \text{clip}\left(\left(\frac{\tau}{\text{PPL}_t + \epsilon}\right)^{\alpha}, \delta_{\min}, \delta_{\max}\right). \quad (7)$$

Here, $\tau$ is a normalization constant, and $\alpha$ controls the steepness of the weighting penalty. This formulation ensures that tokens with probability mass significantly higher than the threshold $\tau$ are down-weighted, effectively regularizing the update steps on sophisticated tokens.

**Objective Function.** We modify the policy's log-likelihood to be a weighted average over the sequence length $T$. The weighted sequence log-probability is given by:

$$\log \tilde{\pi}(y \mid x) = \frac{1}{\sum_{t=1}^{T} w_t} \sum_{t=1}^{T} w_t \log \pi(y_t \mid x, y_{<t}). \quad (8)$$

Substituting this into the standard DPO objective, we obtain the PW-DPO loss:

$$\mathcal{L}_{\text{PW-DPO}}(\pi_\theta; \pi_{\text{ref}}) = -\mathbb{E}_{(x, y_w, y_l) \sim \mathcal{D}}$$
$$\left[\log \sigma\left(\beta\left(\log \frac{\tilde{\pi}_\theta(y_w|x)}{\tilde{\pi}_{\text{ref}}(y_w|x)} - \log \frac{\tilde{\pi}_\theta(y_l|x)}{\tilde{\pi}_{\text{ref}}(y_l|x)}\right)\right)\right].$$

By suppressing the influence of high-perplexity tokens, the objective prioritizes optimization on tokens where the reference model provides reliable support, thereby stabilizing the training process and preventing overfitting to low-probability data patterns. We also propose combining utility-aware preference optimization with perplexity weighting to synergize the strengths of both methodologies. Specifically, we integrate the utility-aware data enhancement mechanism from UAPO into the PW-DPO framework. We term this unified algorithm as PW-UAPO.

# 5. Numerical Experiments

In this section, we present a comprehensive evaluation of the proposed framework for OptProver. We begin by establishing OptBench, a curated benchmark designed to rigorously assess formal reasoning capabilities within the optimization domain. Leveraging this setup, we benchmark OptProver against state-of-the-art theorem provers, providing empirical evidence that expert iteration combined with utility-aware and perplexity-weighted preference optimization significantly enhances the model's proving capabilities. Beyond in-domain accuracy, we verify the model's robustness on general mathematical benchmarks to rule out catastrophic forgetting. Besides, we investigate the scaling laws of expert iteration, and conduct ablation studies to explore the impact of the choices of parameters in PW-DPO.

## 5.1. Benchmark Construction

We introduce OptBench, a novel benchmark comprising 400 undergraduate-level optimization problems curated from the Optlib (Li et al., 2025; 2026). Through a rigorous process of cleaning and extraction, we derive theorems and lemmas from the original repository and categorized them into three primary domains. The first domain, basics, consisting of 121 problems, encompasses fundamental theorems essential for proving optimization properties, such as high-dimensional gradients and Taylor expansion. The second domain focuses on convexity, which contains 135 problems. This part incorporates essential definitions for convex optimization, specifically subgradients, subderivatives, and proximal operators. The third domain involves algorithm analysis, where we extracted partial proofs from numerical algorithm analysis to evaluate whether the model can assist in verifying specific algorithmic steps. This part includes 144 problems. Further details and examples are provided in Appendix B.

## 5.2. Experimental and Evaluation Setup

To ensure the rigor and reproducibility of our results, all experiments are conducted within Lean 4.10.0 environment, fully integrated with LeanDojo v2.1.3 (Yang et al., 2023). Our primary evaluation target is the OptBench dataset constructed in the previous subsection. In addition to this domain-specific benchmark, we also employ the standard MiniF2F-test (Zheng et al., 2021) and ProofNet-test (Azerbayev et al., 2023) benchmarks to monitor the model's general formal proving capabilities beyond optimization tasks.

OptProver is a 7B-parameter model fine-tuned on BFS-Prover-V2 (Xin et al., 2025b), utilizing a best-first search algorithm following the methodology in (Xin et al., 2025a). To demonstrate the effectiveness of our approach, we compare our trained model against state-of-the-art baselines with similar parameter scales, including DeepSeek-Prover-V2-7B (Ren et al., 2025), Goedel-Prover-V2-8B (Lin et al., 2025), and Kimina-Prover-7B (Wang et al., 2025). All whole-proof generation models are tested under Pass@32 and Pass@256. For step-level prover, we use an accumulative budget within Pass@1 and Pass@32.

## 5.3. Capability in Optimization

A clear trend is that step-level proving substantially outperforms whole-proof generation on OptBench. Among whole-proof baselines, the best average performance is below 20% (Pass@256). In contrast, the step-level BFS-Prover-V2-7B already improves to 32.00% Pass@32 on average. Besides, OptProver shows strong optimization capability after two rounds of expert iteration. The full results are reported in Table 2, and we provide case studies in Appendix D.

Specifically, OptProver with expert iteration yields a large

*Table 2.* Performance comparison on OptBench. Whole-proof methods use Pass@32/256, while step-level methods use Pass@1/32. Basic, convex, and algorithmic are the three subcategories, while Avg indicates the average accuracy across the full benchmark.

| Method | Basic | | Convex | | Algorithmic | | Avg | |
|---|---|---|---|---|---|---|---|---|
| | Pass@32 | Pass@256 | Pass@32 | Pass@256 | Pass@32 | Pass@256 | Pass@32 | Pass@256 |
| *Whole-proof Generation Baselines* | | | | | | | | |
| DeepSeek-Prover-V2-7B | 14.88% | 19.01% | 23.70% | 31.85% | 4.86% | 6.25% | 14.25% | 18.75% |
| Goedel-Prover-V2-8B | 14.05% | 19.01% | 19.26% | 32.59% | 4.86% | 8.33% | 12.50% | 19.75% |
| Kimina-Prover-7B | 4.96% | 10.74% | 10.37% | 16.30% | 2.08% | 5.56% | 5.75% | 10.75% |

| Method | Basic | | Convex | | Algorithmic | | Avg | |
|---|---|---|---|---|---|---|---|---|
| | Pass@1 | Pass@32 | Pass@1 | Pass@32 | Pass@1 | Pass@32 | Pass@1 | Pass@32 |
| *Step-level Provers* | | | | | | | | |
| BFS-Prover-V2-7B | 14.05% | 36.36% | 20.00% | 44.44% | 6.25% | 16.67% | 13.25% | 32.00% |
| *OptProver Variants (Ours)* | | | | | | | | |
| OptProver (Only EI) | 32.23% | 49.59% | 39.26% | 60.00% | 15.97% | 40.97% | 28.75% | 50.00% |
| OptProver (EI + DPO) | 34.71% | 52.07% | 39.26% | 62.22% | 20.14% | 42.36% | 31.00% | 52.00% |
| OptProver (EI + UAPO) | 36.36% | 53.72% | **45.93%** | 60.00% | 27.08% | 47.22% | 36.25% | 53.50% |
| OptProver (EI + PW-DPO) | 37.19% | 53.72% | 45.19% | **62.96%** | 27.08% | 46.53% | 36.25% | 54.25% |
| OptProver (EI + PW-UAPO) | **40.50%** | **55.37%** | 45.19% | 62.22% | **28.47%** | **48.61%** | **37.75%** | **55.25%** |

jump of performance over BFS-Prover across all categories, achieving 28.75% Pass@1 and 50.00% Pass@32 on average. This gain is consistent in the three subcategories, basic, convex, and algorithmic, for both Pass@1 and Pass@32, demonstrating that expert iteration alone produces a markedly stronger step policy for optimization problems.

Beyond EI, preference optimization further amplifies performance under the same evaluation budget. Standard DPO already improves the average to 52.00% Pass@32, while our tailored methods deliver larger gains. In particular, utility-aware preference optimization (UAPO) achieves 53.50% Pass@32 by explicitly accounting for the utility of stagnant tactics. Complementarily, PW-DPO reaches 54.25% Pass@32 by reweighting preference updates to better respect the EI-initialized policy, which mitigates catastrophic forgetting and stabilizes optimization of the step policy. Combining both advantages, PW-UAPO achieves the best overall result at 55.25% Pass@32. Notably, the gains are most pronounced on algorithmic and basic problems. In these domains, performance is often degraded by steps that are locally syntactically correct but not helpful to the overall proof. Overall, these results highlight that our utility-aware and perplexity-weighted preference optimization is not merely a drop-in replacement for DPO, but a principled refinement that improves effectiveness and robustness for proof search on optimization-style formal problems.

**Comparison of different kind of provers.** Whole-proof generation remains a significant bottleneck on optimization problems for current 7B-scale proof models. Their performance can only reach 5%–15% Pass@32 and 10%–20% Pass@256, indicating that end-to-end proof generation is

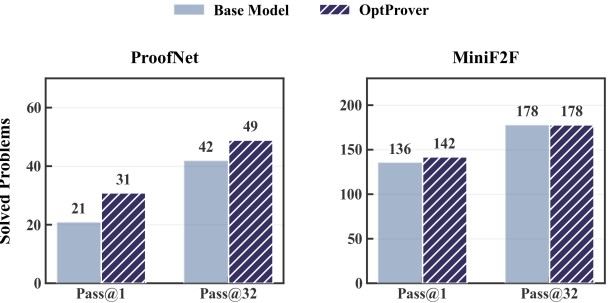

*Figure 3.* OptProver performance on ProofNet and MiniF2F. Base model denotes the original BFS-Prover-V2-7B model. OptProver achieves a clear improvement on ProofNet. On MiniF2F, Pass@1 increases while Pass@k remains unchanged.

highly brittle in this domain. In the whole-proof regime, the model must emit a single verifier-accepted script, which imposes stringent global constraints on lemma selection, library usage, tactic sequencing, and exact syntactic well-formedness. Any mismatch inevitably leads to outright failure. We attribute this brittleness to two compounding factors as follows.

Firstly, optimization benchmarks exhibit a large distribution shift from prior training data, featuring substantial high-dimensional calculus and use of extended real numbers that are underrepresented in existing corpora. While some proof patterns, such as inequality manipulation and standard analytic arguments, transfer across domains, whole-proof generation offers little opportunity to recover once an early choice is misaligned. In contrast, step-level interaction can condition on intermediate proof states to iteratively refine

local decisions and mitigate domain shift.

Secondly, whole-proof generation is particularly sensitive to the correct understanding and deployment of newly introduced or highly specialized definitions. Without feedback, models often fail to unfold, rewrite, or normalize goals into library-aligned forms, preventing effective lemma matching. Step-level settings alleviate this by enabling definition-driven navigation via tactics, such as simplification (`simp` and `dsimp`), definitional unfolding (`unfold`), rewriting (`rw`), which progressively exposes the appropriate definitions and premises needed for proof completion.

### 5.4. Robustness and Out-of-Domain Evaluation

A prevalent risk in domain-specific fine-tuning is catastrophic forgetting, where the model sacrifices general reasoning capabilities to overfit the target distribution. We investigate this trade-off by evaluating our OptProver against the base model on two out-of-domain benchmarks, MiniF2F and ProofNet. As shown in Figure 3, our model demonstrates robustness. On the general MiniF2F benchmark, our model does not exhibit degradation. Instead, it maintains performance parity with the base model on Pass@32 and achieves a slight improvement in Pass@1 accuracy. More notably, performance on ProofNet, which is a general undergraduate benchmark, improves rather than degrades. Our model increases the number of solved problems from 21 to 31 at Pass@1, and from 42 to 49 at Pass@32. This gain suggests that the reasoning patterns learned during our optimization-centric fine-tuning generalize positively to other undergraduate formal mathematical domains. Additional case studies of additional solved problems are provided in Appendix D.3.

### 5.5. Performance of Expert Iteration

We further analyze the scaling behavior of the expert iteration loop across multiple rounds on OptBench and ProofNet. As shown in Figure 4, OptBench performance improves substantially during expert iteration. Both Pass@1 and Pass@32 increase sharply from the base model to the model after the first round of expert iteration. Further rounds continue to improve performance, but the improvements become smaller in each round, indicating diminishing returns. This pattern suggests that EI quickly strengthens performance on easier-to-moderate problems, while later iterations focus increasingly on the remaining hard cases. Progress is limited by the underlying data capabilities and the deeper dependencies in longer proofs, which leave little room for error.

Beyond expert iteration, preference optimization provides an additional source of improvement after EI has largely saturated. In particular, the gain from further EI rounds becomes marginal after the third round of expert iteration.

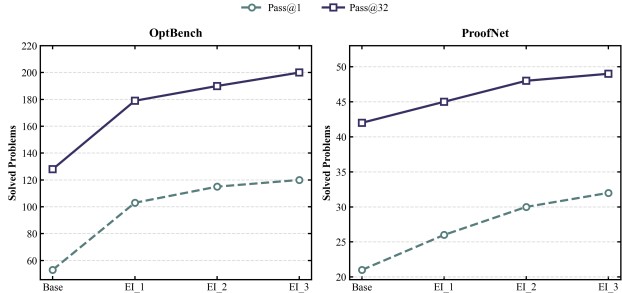

*Figure 4.* Performance of OptProver across expert iteration (EI) rounds on OptBench and ProofNet. The largest gain occurs in the first iteration, followed by diminishing returns in later rounds, suggesting bottlenecks in the remaining hard problems.

*Table 3.* Ablation study of the PW-DPO hyperparameter $\tau$ with accuracy reported on the full OptBench. We set $\tau$ to three representative values from the empirical distribution: $\tau = 1.00$ (25th percentile), $\tau = 1.08$ (median), and $\tau = 2.42$ (75th percentile).

| Parameter Variant | Pass@1 | Pass@32 |
|---|---|---|
| $\tau = 1.00$ | 36.00% | 54.00% |
| $\tau = 1.08$ | **36.25%** | **54.25%** |
| $\tau = 2.42$ | 32.75% | 53.25% |

Under the same hyperparameters, adding a fourth EI round only improves OptBench Avg. Pass@32 from 50.0% to 50.5%. In contrast, applying preference optimization on top of the model after the third round of expert iteration leads to a much larger improvement, with PW-UAPO reaching 55.25% Pass@32. These results suggest that preference optimization serves as a complementary second-stage refinement beyond saturated EI, rather than merely reproducing the effect of additional EI rounds.

### 5.6. Ablation Study

Further ablation study on the normalization constant $\tau$ in the proposed perplexity-weighted DPO method is conducted. $\tau$ controls the normalization of the weighting in (7), effectively setting the reference perplexity scale at which tokens start to be down-weighted. Notably, the perplexity distribution under $\pi_{\text{ref}}$ is heavily skewed towards lower values. $\tau = 1.00$ corresponds to both the minimum and the 25th percentile. As shown in Table 3, choosing $\tau$ to match the median token-level perplexity 1.08 yields the best performance on OptBench, slightly improving over $\tau = 1$. In contrast, setting $\tau$ to a larger value corresponding to the 75th percentile 2.42 degrades performance. This trend supports the role of $\tau$. An overly large $\tau$ weakens the suppression of high-perplexity tokens, allowing out-of-distribution regions to contribute heavily, while a $\tau$ near the central tendency better balances learning signal preservation and stability.

# 6. Conclusion

We introduce OptProver, a specialized formal theorem prover bridging the gap between competition-level mathematics and undergraduate optimization. Our work demonstrates that direct continual training suffers from a significant interactive distribution shift. To address this, we developed a robust pipeline centered on expert iteration and utility-aware preference learning that penalizes stagnant steps, and leverages a perplexity-weighted objective to mitigate catastrophic forgetting. Empirical evaluations on our novel OptBench for optimization reveal that OptProver achieves state-of-the-art performance on optimization, solving over 55% of the problems, while maintaining robust general capabilities on MiniF2F and ProofNet. This work provides a robust framework for domain-specific adaptation in formal theorem proving, thus enabling effective automated theorem proving for optimization problems.

## Impact Statement

This paper presents work whose goal is to advance the field of automatic theorem proving. There are many potential societal consequences of our work, including but not limited to exploring prover models for new specialized mathematical areas. Extending automated reasoning to these domains is a practical step toward ensuring the reliability of LLM systems and reducing the risk of human error in complex mathematical verification.

## Acknowledgments

Z. Wen was supported in part by National Key Research and Development Program of China under the grant number 2024YFA1012903, the National Natural Science Foundation of China under the grant numbers 12331010 and 12288101, and the Natural Science Foundation of Beijing, China under the grant number Z230002. K. Yuan was supported by NSFC No. 12288101.

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

## A. Brief Introduction to Lean

Lean is a general-purpose functional programming language and interactive theorem prover based on dependent type theory. It is primarily designed for formal verification. Mathematical statements are proved by constructing objects that the system can check. Lean relies on a small, trusted kernel that checks every inference step, which offers assumptions and goals for each step. This design has supported the development of Mathlib, a large community-maintained library that covers many areas of mathematics, including algebra, topology, and analysis.

Lean provides an interactive proof development workflow. Although proofs are represented as terms in the underlying type theory, writing proofs directly as terms is often cumbersome. In practice, many users rely on tactic-style proofs, where one works with a proof state consisting of goals together with a local context of variables and hypotheses. Tactics transform the proof state step by step. They introduce assumptions, apply lemmas, rewrite expressions, and decompose a complex goal into smaller subgoals. This style supports incremental proof construction while preserving machine-checked correctness.

Our experiments are built on *Optlib*, a Lean 4 library focused on mathematical optimization. Optlib extends Mathlib with definitions and lemmas commonly used in convex analysis and optimization, including subgradients, tangent cones, and Lagrangian duality. It also provides components for stating and proving first-order optimality conditions and Karush-Kuhn-Tucker (KKT) conditions. In addition, Optlib contains formalized convergence results for several classical optimization algorithms, such as gradient descent, proximal gradient descent and block coordinate descent. These developments make Optlib a suitable base library for the formalization tasks considered in this work.

**LeanDojo**   To connect learning-based methods with interactive theorem proving, we use the system of LeanDojo, an open-source framework that enables programmatic interaction with Lean. LeanDojo allows an external agent to query the current proof state, execute tactics, and obtain feedback from the prover. This interface makes it possible to treat theorem proving as a sequential process in which each step updates the proof state.

LeanDojo also supports extracting training and evaluation data from large libraries such as Mathlib. By instrumenting proof execution, it can produce datasets of state-tactic pairs that record the local context and the active goal at each step. Moreover, LeanDojo provides utilities for working with large collections of available lemmas, which is useful when an agent needs to identify relevant results from the library. In our work, LeanDojo serves as the environment for training and evaluating the agent.

## B. Benchmark Characteristic

Optimization is an inherently multidisciplinary domain that spans a broad spectrum of mathematics, encompassing analysis, linear algebra, set theory, and topology. This makes OptBench different from traditional formal benchmarks such as MiniF2F. We characterize our newly constructed benchmark OptBench from the following aspects.

**High-dimensional Calculus**   Our dataset explicitly includes fundamental theorems in high-dimensional calculus to bridge the gap between elementary scalar math and modern optimization theory. Unlike standard Olympiad-level problems that often focus on specific tricks, the following problem represents the structural reasoning required in general vector spaces. The problem states the mean value theorem in a Hilbert space, and this entry benchmarks the model's ability to handle the rigorous justification of gradient updates, which is the cornerstone of theoretical analysis for optimization.

```
variable {E : Type*} [NormedAddCommGroup E] [InnerProductSpace ℝ E] [CompleteSpace E]

variable {x p y : E} {f : E → ℝ} {f′ : E → E} {s : Set E}

theorem problem_1 (hf : ∀ x : E, HasGradientAt f (f′ x) x) (x p : E) :
  ∃ t : ℝ, t > 0 ∧ t < 1 ∧ f (x + p) = f x + inner (f′ (x + t · p)) p := by sorry
```

**Newly Introduced Definitions**   In addition to reusing standard concepts from Mathlib, OptBench contains many tasks that introduce optimization-specific definitions and ask the model to work with them. For instance, we define subgradients and subderivatives relative to a set $s$. Problem_2 then asks to prove a basic existence statement. Under convexity on $s$ and continuity on $\mathrm{interior}(s)$, every interior point admits at least one subgradient. This tests the ability to reason from newly

introduced definitions rather than relying only on familiar library lemmas.

```
def HasSubgradientWithinAt (f : E → ℝ) (g : E) (s : Set E) (x : E) : Prop :=
 ∀ y ∈ s, f y ≥ f x + inner g (y − x)

def SubderivWithinAt (f : E → ℝ) (s : Set E) (x : E) : Set E :=
 {g : E | HasSubgradientWithinAt f g s x}

variable {E : Type*} [NormedAddCommGroup E] [InnerProductSpace ℝ E] [CompleteSpace E]

variable {f : E → ℝ} {x : E} {s : Set E}

theorem problem_2 (hf : ConvexOn ℝ s f) (hc : ContinuousOn f (interior s)) : ∀ x ∈ interior
    s, (SubderivWithinAt f s x).Nonempty := by sorry
```

OptBench also includes new definitions for algorithmic primitives that commonly appear in modern optimization, such as proximal operators. The predicate `prox_prop` encodes that a point $x_m$ minimizes the proximal objective $u \mapsto f(u) + \|u - x\|^2 / 2$ over `univ`. The theorem `proximal_add_sq` relates proximal problems under a quadratic regularization and a rescaling/shift of both the objective and the center. Such entries reflect the fact that optimization formalizations often require introducing custom predicates and then proving the algebraic and variational properties needed to connect them to standard lemmas. Overall, these problems highlight that OptBench is not only about proving known theorems, but also about working with a large number of newly introduced, domain-specific definitions.

```
variable {E : Type*} [NormedAddCommGroup E] [InnerProductSpace ℝ E] [CompleteSpace E]
    [ProperSpace E]

def prox_prop (f : E → ℝ) (x : E) (xm : E) : Prop :=
 IsMinOn (fun u ↦ f u + ‖u − x‖ ^ 2 / 2) univ xm

variable {x : E} {s : Set E} {f : E → ℝ}

theorem proximal_add_sq (a : E) {l : ℝ} (lpos : 0 < l) (f : E → ℝ):
  ∀ z : E, prox_prop (fun x ↦ f x + l / 2 * ‖x − a‖ ^ 2) x z ↔
  prox_prop ((1 / (l + 1)) · f) ((1 / (l + 1)) · (x + l · a)) z := by
```

## C. Experiment Settings

### C.1. Model and Data

OptProver is initialized from the `BFS-Prover-V2-7B` checkpoint. Our training corpus comprises approximately 60K formalized theorems and 240K state-tactic pairs, constructed from two different sources. The first source consists of formalized versions of two undergraduate textbooks: *Convex Analysis* and *Real Analysis*, which provide foundational theory in optimization-relevant domains. The second source comprises expert iteration trajectories generated through large-scale self-play on `Mathlib` and the formalized `DeepTheorem` dataset.

### C.2. Training and Inference Configuration

The policy model undergoes supervised fine-tuning (SFT) for 2 epochs with a cosine learning rate schedule decaying from $2 \times 10^{-5}$ to $2 \times 10^{-6}$.

During proof search, the system employs a Best-First Search (BFS) algorithm with the following hyperparameters: sampling temperature $\tau = 1.3$ to encourage exploration of diverse tactic sequences, expansion width $k = 4$ to balance search breadth against computational cost, and length normalization factor $\alpha = 2.0$ to mitigate bias toward shorter proof paths.

## C.3. DPO and PW-DPO Hyperparameter Configuration

For the standard Direct Preference Optimization (DPO) method, we set the core regularization coefficient $\beta = 0.01$ to balance the aggressiveness of policy updates and training stability.

For the perplexity-weighted DPO (PW-DPO) method, we adopt the following default hyperparameter configuration: the steepness control parameter $\alpha = 1.0$, the minimum clipping weight $\delta_{\min} = 0.0$, and the maximum clipping weight $\delta_{\max} = 1.0$.

# D. Case Studies

## D.1. Proofs Found by Whole Proof Generation Models of OptBench

Whole proof generation models exhibit limited performance on undergraduate-level optimization problems. Notably, the proofs generated by these models for OptBench problems are significantly shorter than those they produce for olympiad problems, a phenomenon that indicates such models struggle considerably with tackling more challenging undergraduate optimization problems in OptBench.

The following is a proof by DeepSeek-Prover-7B showing that the kernel of any injective linear map is bottom. The prover establishes the conclusion by fully unfolding the definition of the kernel and performing detailed matrix computation.

```
variable {X} [InnerProductSpace ℝ X]
variable [NormedAddCommGroup Y] [InnerProductSpace ℝ Y]
variable (A : X →L[ℝ] Y)

theorem KerA_bot (fullrank: Injective A) : ker A = ⊥ := by
  have h : ker A = ⊥ := by
    apply le_antisymm
    · -- Prove that ker A ⊆ ⊥
      intro x hx
      -- If x is in ker A, then A x = 0, which means x is in the span of the
    zero vector
      have h₁ : A x = 0 := by simpa using hx   -- Since A is injective, x must be the
    zero vector
      have h₂ : x = 0 := by
        have h₃ : A x = A 0 := by simp [h₁]
        have h₄ : x = 0 := by
          apply fullrank
          exact h₃
        exact h₄
      -- Therefore, x is in the span of the zero vector
      simpa [h₂] using zero_mem_bot
    · -- Prove that ⊥ ⊆ ker A
      intro x hx
      -- If x is in the span of the zero vector, then x = 0
      have h₁ : x = 0 := by simpa using hx
      -- Since A is injective, A x = A 0 implies x = 0
      have h₂ : A x = 0 := by
        rw [h₁]
        simp
      -- Therefore, x is in ker A
      simpa using h₂
      -- The final result follows from the antisymmetry of the inclusion
    relation
  exact h
```

Below is the proof generated by Goedel-Prover-V2-8B for a basic property related to matrix. Goedel-Prover unfolds the matrix multiplication and proves the desired property through strict inequality analysis.

```
open scoped BigOperators
open Matrix

variable {m n : ℕ}

theorem Transpose_mul_self_eq_zero {A : Matrix (Fin m) (Fin n) ℝ} :
  Aᵀ * A = 0 ↔ A = 0 := by
 have h_forward : Aᵀ * A = 0 → A = 0 := by
  intro h
  have h₁ : ∀ i j, A i j = 0 := by
   intro i j
   have h₂ : (Aᵀ * A) j j = 0 := by
    rw [h]
    exact Matrix.zero_apply j j
   have h₃ : (Aᵀ * A) j j = ∑ k : Fin m, A k j * A k j := by
    simp [Matrix.mul_apply, Matrix.transpose_apply, Fin.sum_univ_succ]
    <;>
    rw [Fin.sum_comm]
   rw [h₃] at h₂
   have h₄ : ∑ k : Fin m, A k j * A k j = 0 := h₂
   have h₅ : A i j = 0 := by
    by_contra h₅
    have h₆ : A i j ≠ 0 := h₅
    have h₇ : (A i j)^2 > 0 := by
     exact sq_pos_of_ne_zero h₆
    have h₈ : ∑ k : Fin m, A k j * A k j ≥ (A i j)^2 := by
     have h₉ : A i j * A i j ≤ ∑ k : Fin m, A k j * A k j := by
      exact Finset.single_le_sum
       (fun k _ => mul_self_nonneg (A k j)) (Finset.mem_univ i)
     nlinarith
    have h₉ : ∑ k : Fin m, A k j * A k j > 0 := by
     linarith
    linarith
   exact h₅
  have h₂ : A = 0 := by
   ext i j
   exact h₁ i j
  exact h
 have h_backward : A = 0 → Aᵀ * A = 0 := by
  intro h
  rw [h]
  simp [Matrix.mul_zero]
  <;>
  simp_all [Matrix.zero_mul]
  <;>
  aesop
 have h_main : Aᵀ * A = 0 ↔ A = 0 := by
  constructor
  · intro h
   exact h_forward h
  · intro h
   exact h_backward h
 exact h_main
```

```
variable {E : Type*} [NormedAddCommGroup E] {xm : E} {f : E → ℝ} {g : ℕ → E}
open Set Finset
```

```
theorem mono_sum_prop_primal (mono : ∀ k : ℕ, f (g (k + 1)) ≤ f (g k)):
  ∀ n : ℕ, (Finset.range (n + 1)).sum (fun k ↦ f (g (k + 1))) ≥ (n + (1 : ℝ)) * f (g (n + 2)) := by
  intro n
  induction n with
  | zero =>
    norm_num
    <;> simp_all [Finset.sum_range_succ]
    <;> linarith [mono 0, mono 1]
  | succ n ih =>
    simp_all [Finset.sum_range_succ, Nat.cast_add, Nat.cast_one, add_mul, mul_add,
      mul_one]
    <;> nlinarith [mono (n + 1), mono (n + 2), mono (n + 3), ih]
```

## D.2. Proofs Found by OptProver of OptBench

**Proofs of Hard Problems found by OptProver.**  Our OptProver can solve some hard problems that cannot be solved by other whole proof generation models or the original BFS-Prover-V2-7B. We give some case studies here.

The following problem is extracted from the convergence analysis proof of the alternating direction method of multipliers (ADMM). OptProver leverages the assumptions in the given setting to prove a complex inequality related to the update iteration.

```
lemma problem_example₁ [Setting E₁ E₂ F admm admm_kkt]: ∀ n : ℕ+,
  (min τ (1 + τ − τ ^ 2)) * ρ * ‖A₂ (x₂ n − x₂ (n + 1))‖ ^ 2
  + (min 1 (1 + 1 / τ − τ)) * ρ * ‖A₁ (e₁ (n + 1)) + A₂ (e₂ (n + 1))‖ ^ 2
  ≥ 0 := by
  intros n
  apply add_nonneg
  apply mul_nonneg
  apply mul_nonneg
  pick_goal 4
  apply mul_nonneg
  apply mul_nonneg
  rotate_left
  linarith [admm.hrho]
  · apply sq_nonneg
  pick_goal 3
  apply sq_nonneg
  rw [min_def]
  rotate_left
  repeat' apply le_of_lt
  · exact admm.hrho
  all_goals aesop
  rw [← sub_pos]
  rw[sub_eq_add_neg]
  by_contra h
  apply h
  rotate_left 1
  all_goals contrapose! h
  revert h
  intro hτ
  rw [← sub_pos]
  rw[add_assoc]
  rotate_left
  on_goal −1 => revert h
  have := hτ
  have : (1 : ℝ) ≤ τ + (1 − τ) ^ 2 := by nlinarith
  any_goals aesop
  any_goals contrapose this
```

```
any_goals push_neg at *
rw [← sub_pos]
rw [sub_eq_add_neg]
on_goal −1 => rw [add_comm]
any_goals contrapose this_1
on_goal 1 => revert this_1
simp
intro h
pick_goal 3
rw[add_assoc]
rw [add_comm]
rotate_left
all_goals contrapose h
on_goal 2 => aesop
simp at h ⊢
pick_goal 3
apply not_le_of_gt
any_goals contrapose h
on_goal −1 => revert h
any_goals simp_all
rotate_left
rw [add_comm] at h
rotate_left
rw [pow_two]
intros
rotate_left
rw [add_assoc]
rotate_left
all_goals rcases admm.htau with ⟨h_1, _⟩
positivity
all_goals ring
all_goals aesop (add simp [pow_two])
any_goals field_simp at *
on_goal 1 => rw [← sub_le_iff_le_add′]
apply sub_le_sub_left
on_goal 2 => rw [← sub_pos]
on_goal 2 =>
 first | nlinarith | rw [← sub_pos]
on_goal 1 => contrapose h
rotate_left
simp only [sub_zero]
apply by_contra
intro h
push_neg at h
apply h_1.not_le
apply le_of_add_le_of_nonneg_left
on_goal 1 => rw [_root_.add_comm]
rotate_left
· exact 1 − τ
all_goals contrapose h
any_goals contrapose h
any_goals ring_nf at *
simp only [not_not]
rotate_left
simp only [not_not] at *
any_goals contrapose h
any_goals simp at *
rw[abs_of_pos h_1]
any_goals contrapose right
all_goals field_simp at *
```

```
repeat′ nlinarith [Real.sq_sqrt (show 0 ≤ 5 by norm_num),
 Real.sqrt_nonneg 5]
contrapose! right
rw [← sub_pos] at right
set a := (1 + Real.sqrt 5) / 2
revert this
intro H
have h₂ := sq_pos_of_pos h₁
rw [add_comm]
rw [sub_eq_add_neg]
rw [← sub_pos]
set x := τ
norm_num at *
simp [(sq x), mul_assoc]
rw [div_le_iff (by positivity)] at H
rw [add_comm, mul_comm] at H
apply add_pos_of_pos_of_nonneg
rotate_left
any_goals nlinarith [sq_nonneg (x − 1), Real.sq_sqrt (show 0 ≤ 5 from by norm_num),
 Real.sqrt_pos.2 (show (0 : ℝ) < 5 from by norm_num), mul_pos h₁ (Real.sqrt_pos.2 (by
    norm_num : (0 : ℝ) < 5))]
rw [show x + −(x * x) = x * (1 − x) by ring]
refine mul_pos h₁ (sub_pos.mpr ?_)
cases′ lt_or_le 1 x with h″ h″
contrapose! H
any_goals contrapose right
all_goals field_simp [a]
all_goals contrapose right ; ring_nf at *
any_goals
 push_neg at *
 nlinarith [Real.sq_sqrt (show 0 ≤ 5 by positivity)]
```

The proof requires more than one hundred lines, primarily due to the strictness of formal language in expressing inequalities—details that are often omitted in natural language proofs.

Another example is about the proof of the strongly convex property. This definition rarely appears in high-school-level olympiad problems. OptProver can address such problems after being trained on the optimization-focused data corpora.

```
variable {E : Type*} [NormedAddCommGroup E] [InnerProductSpace ℝ E] [CompleteSpace E]
variable [ProperSpace E]

lemma strongconvex_of_convex_add_sq (f : E → ℝ) (x : E) (hfun : ConvexOn ℝ univ f) :
  StrongConvexOn univ (1 : ℝ) fun u ↦ f u + ‖u − x‖^2 / 2 := by
rw [StrongConvexOn]
simp_rw [add_comm (f _)]
constructor
· exact convex_univ
rintro y − z − a b ha hb hab
simp only [smul_eq_mul, sub_add, div_eq_inv_mul]
ring_nf
replace hfun := hfun.2
specialize hfun (by simp) (by simp) ha hb hab
exact y
· exact z
simp only [_root_.div_eq_inv_mul] at hfun ⊢
field_simp
refine (div_le_div_right two_pos).2 ?_
rw [norm_sub_sq_real, norm_sub_sq_real, norm_sub_sq_real]
rcases eq_sub_of_add_eq hab with rfl
simp [norm_add_sq_real, inner_add_left, inner_smul_left]
```

```
rw [norm_smul, norm_smul, inner_smul_right, real_inner_comm]
simp only [Real.norm_eq_abs, abs_of_nonneg hb, abs_of_nonneg ha, sub_mul, sub_add,
 add_sub_assoc, mul_assoc]
norm_num
on_goal 1 => norm_num at *
have : 0 ≤ 1 - b := by linarith
conv_lhs => rw [add_comm, add_assoc, add_left_comm]
ring_nf
norm_num [real_inner_comm] at *
ring_nf at *
repeat' rw [norm_sub_sq_real]
rw [mul_comm b] at hfun
nlinarith [real_inner_comm x y, real_inner_comm x z, real_inner_comm y z,
 sq_nonneg (b - 1), sq_nonneg (b - 2), f y, f z]
```

**Techniques Used in Step-level Proofs**    In step-level proving, domain-specific definitions such as the 'Lagrange_function' and 'KKT_point' are commonly introduced. To reason reliably about these novel abstractions, step-level provers explicitly unfold such definitions through definitional rewriting and simplification—a process that unveils their underlying concrete properties, including feasibility conditions, multiplier nonnegativity, and gradient-based statements, which would otherwise remain concealed behind high-level terminology. Once definitions are unfolded, provers can manipulate explicit equalities and inequalities in place of opaque predicates, enabling straightforward rewriting and simplification steps in subsequent reasoning. This "unfold-then-reason" paradigm reduces reliance on domain-specific heuristics and significantly enhances transferability to unfamiliar definitions and problem formulations in optimization.

```
variable {E : Type _} {τ σ : Finset ℕ}
variable [NormedAddCommGroup E] [InnerProductSpace ℝ E] [CompleteSpace E]
variable {p : Constrained_OptimizationProblem E τ σ}

theorem KKT_multipliers_objective_eq_Lagrangian {p :
    Constrained_OptimizationProblem E τ σ} (x : E) (lambda1 : τ → ℝ) (lambda2 : σ → ℝ)
 (hKKT : KKT_point p x lambda1 lambda2) :
 p.objective x = p.Lagrange_function x lambda1 lambda2 := by
rw [KKT_point] at hKKT
dsimp [Lagrange_function]
obtain ⟨hgradient, _, lambda_nonneg, _⟩ := hKKT
dsimp only [FeasSet, FeasPoint] at *
simp [*]
nth_rewrite 1 [← sub_zero (p.objective x)]
congr
rw [eq_comm, Finset.sum_eq_zero_iff_of_nonneg]
all_goals (intro i _; simp_all)
```

**Comparison with Whole Proof Generation Models.**    The following is the proof found by OptProver for the problem KerA_bot, which we present the proof found by the whole proof generation model in Appendix D.1. In contrast to DeepSeek-Prover-7B's proof, which involves a detailed discussion of injective linear operator properties, OptProver directly uses a Mathlib theorem to handle the problem.

```
variable {X} [InnerProductSpace ℝ X]
variable [NormedAddCommGroup Y] [InnerProductSpace ℝ Y]
variable (A : X →L[ℝ] Y)

theorem KerA_bot (fullrank: Injective A) : ker A = ⊥ := by
 exact LinearMap.ker_eq_bot_of_injective fullrank
```

The following is the proof found by OptProver for the problem Transpose_mul_self_eq_zero, which we present the

proof found by the whole proof generation model in Appendix D.1. Compared to the complicated method provided by Goedel-Prover, OptProver finds a more concise way to prove it, leveraging theorems in Mathlib.

```
open scoped BigOperators
open Matrix

variable {m n : ℕ}

theorem Transpose_mul_self_eq_zero {A : Matrix (Fin m) (Fin n) ℝ} :
  Aᵀ * A = 0 ↔ A = 0 := by
constructor
intro hA
rw [← Matrix.ext_iff] at hA ⊢
intro i j
specialize hA j j
simp [Matrix.mul_apply] at hA
rw [Finset.sum_eq_zero_iff_of_nonneg] at hA
any_goals aesop (add simp [Matrix.ext_iff])
exact mul_self_nonneg _
```

**Ability in Dealing with Extended Real Numbers.** Extended real numbers is frequently used in optimization. Our OptBench therefore includes several problems related to the calculation rules of the extended real numbers. While such problems seem straightforward, they are far more difficult to solve than traditional real-number inequality problems. This is due to the corner cases posed by $0$ and $\infty$. The whole proof generation models fail to complete the following basic property about extended real numbers.

```
lemma biSup_const_add {E} {a : EReal} {f : E → EReal} {s : Set E}
  (pf : ∀ m ∈ s, f m > ⊥) (pa : a ≠ ⊥) :
  a + (⨆ m ∈ s, f m) = (⨆ m ∈ s, a + f m) := by
induction a
· contradiction
swap
on_goal 1 => rcases s.eq_empty_or_nonempty with (rfl | hne)
· simp
cases isEmpty_or_nonempty E
· rw [iSup_of_empty, iSup_of_empty]
  simp
rotate_left
rw [EReal.biSup_coe_const_add]
simp_all
erw [top_add_of_ne_bot]
rotate_left
rw [ne_eq, iSup_eq_bot]
push_neg
inhabit E
use hne.some
rw [iSup]
any_goals aesop
exact hne.choose_spec
specialize pf _ hne.some_mem
simp [a] at pf
symm
refine biSup_eq_top ?_
refine ⟨hne.some, hne.choose_spec, ?_⟩
apply top_add_of_ne_bot
exact (pf _ (hne.choose_spec)).ne'
```

### D.3. Proof Found by OptProver of ProofNet

We additionally solve extra problems from ProofNet using OptProver. The following is an example. Although this theorem is not specific to real analysis, it is representative of the kind of foundational mathematics. It requires substantial reasoning about cardinality, relating finiteness/infiniteness to the existence of injective maps. Our model succeeds on this instance largely because it has been exposed to related patterns in our real analysis corpus. This supports a way for further model ability development. Improving performance and transfer on heterogeneous benchmarks benefits from widening the data coverage across mathematical areas, together with a stable training procedure that can reliably internalize and reuse such recurring proof motifs.

```
import Mathlib

open Fintype Subgroup Set Polynomial Ideal
open scoped BigOperators
noncomputable section

theorem DummitFoote_exercise_1_3_8 : Infinite (Equiv.Perm ℕ) := by
  inhabit Nat
  haveI infinite_nat := Infinite.of_injective (fun n => n + 1) Nat.succ_injective
  constructor
  contrapose! infinite_nat
  refine' not_infinite_iff_finite.mpr _
  contrapose! infinite_nat
  contrapose infinite_nat
  any_goals simp_all
  contrapose infinite_nat
  intro h
  contrapose! infinite_nat
  simp at *
  contrapose! h
  contrapose h
  simp? at h says simp only [not_not] at h
  simp_all
  contrapose! h
  intro h'
  by_contra contra
  contrapose! contra
  have := h'
  contrapose this
  simp at this ⊢
  constructor
  contrapose! h'
  rw [@not_finite_iff_infinite]
  contrapose! h'
  contrapose h'
  simp at h' ⊢
  revert h'
  contrapose
  rintro hInfinite
  simp_all only [not_false_iff, true_and]
  contrapose! hInfinite
  apply Infinite.of_injective (fun n ↦ (Equiv.swap 0 n : Equiv.Perm ℕ))
  intro a b h
  contrapose! h
  intro heq
  apply h
  rw [@Equiv.ext_iff] at heq
  specialize heq a
  simp only [Equiv.swap_apply_right] at heq
```

```
dsimp [Equiv.swap] at heq
contrapose! heq
intro heq′
symm at heq′
simp [Equiv.swapCore] at heq′
split_ifs at heq′ <;> aesop
```

