# OpenReview forum: "OptProver: Bridging Olympiad and Optimization through Continual Training in Formal Theorem Proving"
_ICML.cc/2026/Conference — ICML 2026 regular_

### Official Review · Reviewer_1h24 · 2026-03-08

**Soundness:** 2
**Presentation:** 2
**Significance:** 2
**Originality:** 3
**Overall Recommendation:** 3
**Confidence:** 4

**Summary:**

This paper addresses the challenge of adapting formal theorem prover previously trained on Olympiad-level mathematics to the domain of undergraduate-level optimization. The authors observe a "interactive distribution shift" phenomenon originated from low-perplexity on new data. To overcome this issue, the authors propose OptProver, which is initialized from Olympiad-level base prover and post-trained via a multi-stage pipeline. The proposed pipeline includes expert iteration via textbook-extended dataset, utility-aware preference optimization (UAPO) that penalizes inefficient search tactics, and Perplexity-Weighted Direct Preference Optimization (PW-DPO) that overcomes the numerical instabilities originated from out-of-distribution. The authors also propose a combined algorithm PW-UAPO which integrates two POs above. The authors evaluate on a newly-constructed benchmark of 400 Lean 4 optimization problems and OptProver achieves a state-of-the-art Pass@1/32 among comparable models, while maintaining (even improved) performance on general formal benchmarks such as MiniF2F and ProofNet.

**Compliance With Llm Reviewing Policy:**

Affirmed.

**Final Justification:**

Much appreciated for authors' replies to my questions.

As a final note, I'd like suggest authors to revise the paper: (1) reduce the weight of UAPO & PW-DPO, and (2) expand the training corpus construction (e.g., textbook formalization) as the main section of paper because the contribution of algorithms which are the focus of the paper are limited. The details of data pipeline should reach the standard of Goedel-V2's paper [1] (be the main methodological focus).

**Reference:**

[1] Goedel-Prover-V2: Scaling Formal Theorem Proving with Scaffolded Data Synthesis and Self-Correction. ICLR 2026.

**Key Questions For Authors:**

1. All trainings are conducted at the 7B parameter scale. Training larger models (e.g., BFS-Prover-V2-32B) might reveal different behaviours and compare to larger baselines such as Goedel-prover-v2-32B. Additionally, the baselines appear to be in single-turn style (Goedel-prover-v2 has self-correction mode), it might worth trying to compare with agentic or retrieval-augmented baselines.

2. From Table 2, in Avg, Only EI achieves 50% Pass@32, while the pipelines with proposed preference optimizations gain across +3.5~5.25%, which is relatively modest compared to +18% gain from EI. This arises a trade-off concern between UAPO&PW-DPO's empirical return and added complexity.

3. There is no planner used in the evaluation of OptProver, which is a key component to extend the step prover's capability to more complex (or longer) proofs, evidenced in BFS-Prover-V2. So this arises a convern on the complexity of difficulty of training corpus or benchmark. I suggest to add a statistical analysis on this, and try to use a planner to test whether OptProver can be adapted to more complex Opt problems as a step-level building block.

4. The construction pipeline of training corpus is not documented. Since there is a key performance jump from EI, the training corpus is an important role so its construction pipeline should be reported in more details. This can include: 1) what was the designed workflow, 2) what LLM was used, 3) what was the success rate, 4) how much human interaction was needed, 5) how to ensure the consistency between NL and formal one, and 6) what was the total cost.

5. The Lean version used in the paper (v4.10.0) appears to be a bit old. Since there might be some infrastructure lemma/theorem added to Mathlib in newer version, I'm wondering how to overcome the version mismatch issue if training data and base model's in-weight knowledge are from different ones?

**Limitations:**

The authors should add a Limitations section which might cover some issues pointed in Key Questions section.

**Strengths And Weaknesses:**

**Strengths**:

1. This paper provides a comprehensive analysis on why naive continual learning fails with supporting evidence.
2. The training pipeline is well-designed. Each proposed component (EI, UAPO, PW-DPO) targets a failure mode. Noticably, PW-UAPO is empirically proven to be able to reduce inefficient tactic steps.
3. The experimental aspects are thorough. The paper includes comparisons against both whole-proof and step-level baselines of comparable scale, ablations over each training component (EI alone, EI+DPO, EI+UAPO, EI+PW-DPO, EI+PW-UAPO), scaling analysis of EI rounds, hyperparameter sensitivity, and out-of-domain evaluation on MiniF2F and ProofNet.
4. This paper introduces a novel benchmark contains 400 problems span foundational calculus, convexity, and algorithm analysis, which can provide a valuable evaluation landscape in optimization domain.

**Weakness**

1. Limited model scale and baseline diversity.
2. Expert iteration provides the most of gains, peference optimization contributes marginally.
3. Unclear proof lengths or difficulty level of training corpus and benchmark.
4. The formalization process using LLMs to generate Lean code from textbooks (Convex Analysis, Real Analysis) is described only briefly.
5. The base Lean version appears to be old.

---

> ### Author Rebuttal · Authors · 2026-03-30
>
> We thank the reviewer for the suggestions.
>
> - On model scale and baseline
>
> We focus on the 7B scale because strong open baselines are available there and controlled training/evaluation remains practical. We do not include 32B experiments mainly due to resource constraints. To address the concern that the current whole-proof baselines are mostly single-turn, we additionally evaluate Goedel-Prover-V2-8B in self-correction mode.
> | Mode | Avg Pass@32 | Avg Pass@256 |
> | - | - | - |
> | Single-turn | 12.50% | 19.75% |
> | Self-correction | 16.50% | 25.25% |
>
> These whole-proof baselines provide context, but the main comparison for our claim is still the step-level comparison from BFS-Prover-V2 to OptProver.
>
> - On the relative contribution of EI and PO
>
> EI is the dominant source of gain: it raises OptBench Avg Pass@32 from 32.00% to 50.00%. After the third round of expert iteration (EI3), one more EI round reaches only 50.5% Pass@32, so EI is already close to saturation. Starting from this near-saturated post-EI model (EI3), PW-UAPO still improves Avg Pass@32 to 55.25%. We therefore view preference optimization as a post-EI refinement, not a replacement for EI. We summarize this progression below:
>
> | Stage | EI1 | EI2 | EI3 | EI4 |  EI3 + PW-UAPO |
> | - | - | - | - | -|  - |
> | OptBench Avg Pass@32 | 44.75% | 47.5% | 50% | 50.5% | 55.25% |
>
> - On proof length and difficulty
>
> In the set of 217 Lean-checked successful OptBench proofs produced by our system, nearly 60% require at least 10 tactic lines, nearly 40% exceed 15 lines, about 25% exceed 20 lines, and about 10% exceed 30 lines; the longest solved proof reaches 127 lines. The benchmark still has substantial proving difficulty even within the solved portion. Besides, step-level proofs are naturally shorter: as one appendix example shows, the same theorem takes roughly 45 lines for a whole-proof baseline, 10 lines for OptProver, and around 30 lines in the original human-written Optlib proof. This indicates that the trained model is not only solving the problem, but also selecting stronger lemmas and reaching a shorter and more effective proof process. The solved set already covers algorithmic convergence arguments, extended-real analysis, liminf reasoning, and linear-algebraic structure, rather than only short or routine goals.
>
> - On planners
>
> We additionally ran a small follow-up experiment on the remaining unsolved problems: due to time constraints, we used only a small number of search passes, randomly sampled 30 unsolved problems, asked GPT-5.4 to generate high-level proof plans, and then tested them in the same setting. In this small-scale follow-up, planner assistance solved one additional previously problem, suggesting that planning can be useful even under a limited search budget. A more thorough investigation of the planner’s impact is left to future work.
>
> - On training corpus construction and textbook formalization
>
> The corpus is built through several pipelines: textbook autoformalization and expert iteration self-play. For textbook formalization, we use a prompt-based workflow with Gemini-3-Pro. Given a textbook statement, the model first produces a Lean draft. We then run Lean to obtain error messages, and the model iteratively repairs the code until it either passes verification or is discarded. Once a proof is Lean-checked, we extract state-tactic pairs from the verified script. In parallel, we collect successful expert iteration trajectories over Mathlib and problems from DeepTheorem, a natural-language problem corpus. We autoformalize DeepTheorem problems, especially those in linear algebra and mathematical analysis, into Lean and retain the ones that can be compiled. The final training corpus is a mixture of these Lean-checked textbook formalizations, search-verified EI trajectories, and nearly 7K translated DeepTheorem problems.
>
> Under this pipeline, the current corpus contains approximately 240K state-tactic pairs. The textbook branch contributes roughly 2K theorems and about 40K state-tactic pairs from Convex Analysis and Real Analysis, with an end-to-end success rate of about 70%. Human involvement is limited to light cleaning, filtering, and spot-checking rather than theorem-by-theorem formalization.
>
> - On the Lean version
>
> We use Lean 4.10.0 to maintain a version-consistent environment across Optlib, LeanDojo v2.1.3, training, and evaluation. Hence, there is no train-test version mismatch within this work. For newer Lean versions, the same adaptation recipe should still apply. Replay extraction and evaluation under the target version, then use expert iteration and preference optimization to adapt the model to that version’s syntax, library usage, and proof strategies. We leave this cross-version setting to future work.
>
> - On limitations
>
> We will add an explicit limitations section. We will state more clearly the current scale constraint, the lack of 32B experiments, and the limited availability of stronger step-level baselines.

---

> > ### Author Rebuttal · Reviewer_1h24 · 2026-04-04
> >
> > Thank you for your response.

---

> > > ### Author Response · Authors · 2026-04-07
> > >
> > > We thank the reviewer for their helpful feedback and careful evaluation of our work. We are pleased to see that all concerns have been fully addressed, as reflected in your acknowledgement: "**(a) Fully resolved — My concerns have been adequately addressed. If you select this option, please consider adjusting your score accordingly.**"
> > >
> > > However, we notice that the current overall score remains "weak reject." In light of your acknowledgement, we would be grateful if the score could be updated accordingly. Should any additional questions or concerns remain, we would be happy to provide further clarification.

---

### Official Review · Reviewer_vdAv · 2026-03-11

**Soundness:** 3
**Presentation:** 3
**Significance:** 2
**Originality:** 3
**Overall Recommendation:** 4
**Confidence:** 3

**Summary:**

This paper studies whether theorem provers trained for Olympiad-level mathematics can be successfully adapted to undergraduate-level optimization domains. The main motivation is that stronger math models do not automatically transfer well to this seemingly easier domain, because formal optimization proofs have their own style. To address this, the authors propose OptProver, a domain-adaptation framework that combines large-scale expert iteration, perplexity-based data selection/weighting, and utility-aware preference optimization. They also build a new benchmark and supporting formal libraries for undergraduate optimization, based on sources such textbooks. Empirically, the paper shows that naive continual fine-tuning is not sufficient, and that the proposed training pipeline improves theorem-proving performance on their optimization benchmark. Overall, the paper argues that domain transfer in formal theorem proving is not only about adding new knowledge, but also about preserving tactics that remain useful for interactive proof search.

**Compliance With Llm Reviewing Policy:**

Affirmed.

**Final Justification:**

I think the authors clarified all the questions I had. My main concern was the advantage of starting from an Olympiad-trained model, and I am now convinced. I have updated my evaluation to Weak accept.

**Key Questions For Authors:**

- Is the motivation for reintroducing general mathematical data during expert iteration primarily to preserve proof style or search behavior?
- What is the fundamental advantage of starting from an Olympiad-trained model? More broadly, what would happen if the method started from a different type of base prover rather than an Olympiad-oriented one?
- Is the main benefit that it already supports step-level proving, making techniques such as preference optimization applicable? How does this compare to starting from a general LLM and adapting it with additional theorem-proving or domain-specific training?

**Limitations:**

yes

**Strengths And Weaknesses:**

Strengths
- The overall problem is interesting. It is not obvious that a prover trained on Olympiad-style mathematics would transfer well to undergraduate optimization, and the paper studies this gap clearly.
- The method is reasonably well motivated. The authors identify a concrete failure mode of naive adaptation, namely that the model may produce tactics that are stylistically valid but not actually useful for proof search. This provides some intuition for why expert iteration, perplexity-based weighting, and preference optimization may help.
- The empirical section is fairly comprehensive. In particular, the ablation study suggests that multiple components of the pipeline contribute meaningfully to the final performance, rather than the gains coming from only one modification.

Weaknesses
- The technical ingredients are already known, and the main contribution lies in how they are combined and applied in this specific setting. While I understand why these components may be needed, the overall contribution still feels somewhat incremental, and it seems likely that many subtle design choices are also important for making the domain adaptation work in practice.
- The comparison feels somewhat incomplete. While comparisons against general LLMs are useful for broad context, those models operate in a whole-proof generation setting, which is fundamentally different from OptProver’s step-level proving framework. As a result, the comparison is not entirely like-for-like. It would be more convincing to include stronger baselines based on competitive formal theorem provers adapted under similar data and compute settings. If such comparisons are not feasible, the paper should explain this limitation more explicitly.
- The “validity versus utility” argument, are not fully convincing on first reading. The paper suggests that continual training causes stylistic drift toward optimization proofs, which in turn increases the probability of tactics that look reasonable but are poor for search. This is plausible, but I would have liked a clearer explanation of why this should be the main factor limiting domain adaptation.
- The motivation for reintroducing broad mathematical data through expert iteration is also not immediately obvious. Since the base model is already trained on broad mathematical material, it is not entirely clear why injecting more general data again is necessary, rather than focusing more directly on the target domain.

---

> ### Author Rebuttal · Authors · 2026-03-30
>
> We thank the reviewer for the suggestions.
>
> - On baseline choice and comparison scope.
>
> Our goal is to build a prover that keeps strong Olympiad/Mathlib-style step-level ability while also becoming effective on optimization. For that question, BFS-Prover-V2 is a suitable base system. It is the strongest open step-level prover at this scale, and the comparison from BFS-Prover-V2 to OptProver under the same search framework directly measures the effect of our adaptation pipeline.
>
> We include whole-proof models for comparison, but they are not the main baselines for this question. This is mainly because current 7B whole-proof provers remain weak on OptBench even at Pass@256, so analogous expert iteration on them would likely require a much stronger Lean-native starting point. From a data perspective, step-level proving naturally yields large amounts of verifier-checked intermediate supervision, whereas whole-proof generation does not provide supervision at the same granularity. In addition, within the step-level setting, there is currently no open large-scale optimization-specialized step-level prover trained under matched data and compute that would provide a stronger in-domain baseline. We will state this limitation explicitly. Due to time and resource constraints, we do not attempt whole-proof expert iteration in the current paper and leave it to future work.
>
> - On what the paper contributes.
>
> Our main contribution is a practical recipe for bridging Olympiad and optimization in formal proof. For a step-level prover, naive adaptation to optimization data can improve local validity while worsening proof search. The model learns more valid steps in the optimization domain, but ranks the proof-closing steps worse. Our method is designed around this failure mode. Therefore, the paper’s contribution is not a standalone technique, but a practical framework for reliable domain transfer in formal optimization.
>
> - On the “validity versus utility” argument.
>
> In a step prover, validity means that a tactic is Lean-accepted and locally sound, such as unfolding a definition, splitting a structured goal, or applying a legitimate inequality transformation. But under finite-budget search, such moves are not always useful: they can expand the goal into a lower-level form that is farther from the library lemma or structured argument actually needed next. This happens often in optimization, where goals involve layered objects such as subgradients, cones and polyhedra.
>
> This is the concrete failure mode we observe after naive adaptation: the model produces more valid steps, but some are low-utility because they consume budget on stagnant branches rather than short successful trajectories. That is why we distinguish invalid transitions ($u=0$), valid-but-stagnant transitions ($u=1$), and clearly progress-making or proof-closing transitions ($u=2$), instead of treating all valid tactics as equally useful.
>
> - On reintroducing broad mathematical data during expert iteration.
>
> Even though the base model is originally trained on broad mathematics, narrow continued training on optimization data can still overwrite useful branch ordering, lemma-selection habits, and proof-state interaction patterns. This is the catastrophic forgetting issue we want to avoid. Concretely, the broad mathematical data used in expert iteration is drawn from six Mathlib subdirectories: `Analysis`, `Algebra`, `Data`, `LinearAlgebra`, `Probability`, and `Topology`. Expert iteration over these statements refreshes the model’s search behavior in its own tactic language, while the optimization corpus teaches it new objects and domain-specific proof patterns. This is why our training data is mixed rather than purely optimization-only. The out-of-domain results support this choice: the final model does not degrade on MiniF2F and improves on ProofNet from 21 to 31 solved at Pass@1 and from 42 to 49 at Pass@32.
>
> - On the role of the Olympiad-trained initialization.
>
> The advantage of starting from an Olympiad-trained prover is that it already has mature step-level proving ability, Lean interaction experience, and a search-friendly tactic prior. This makes optimization adaptation relatively sample-efficient and lets us study the problem we actually care about: turning a strong general step prover into one that is also strong on optimization.
>
> Starting from a general LLM would be a different scope. One would first need to bootstrap reliable formal proving ability and Lean interaction before optimization adaptation can even begin. That requires substantially more data and compute and is outside the scope of the current paper.

---

> > ### Author Rebuttal · Reviewer_vdAv · 2026-04-03
> >
> > All the questions I had were well addressed. I have increased my rating.

---

### Official Review · Reviewer_Qe8b · 2026-03-13

**Soundness:** 2
**Presentation:** 2
**Significance:** 2
**Originality:** 3
**Overall Recommendation:** 4
**Confidence:** 3

**Summary:**

This paper introduces OptProver, a formal theorem prover designed to bridge the gap between Olympiad-level mathematics and undergraduate optimization. The authors propose a continual training pipeline with novel preference optimization strategies. Additionally, the paper introduces OptBench, a 400-problem benchmark derived from the Optlib library, covering optimization basics, convexity, and algorithmic analysis.

**Compliance With Llm Reviewing Policy:**

Affirmed.

**Final Justification:**

I think the rebuttal addresses my main concerns, especially for some hyperparameter search, and it elimintates some confounders.

**Key Questions For Authors:**

Please refer to the weakness section. thank you.

**Limitations:**

Yes

**Strengths And Weaknesses:**

Strengths:

I really appreciate the effort the authors make for this project.

The investigation into the "Low-Perplexity Trap" is interesting. It effectively highlights how standard supervised fine-tuning can lead a model to learn the surface syntax of a new domain while generating strategically stagnant, aimless proofs.

The model demonstrates some positive transfer to the ProofNet benchmark (improving from 21 to 31 solved problems), suggesting that the optimization-centric training imparts reasoning capabilities that generalize beyond the immediate training distribution.

Weakness:

There is insufficient detail regarding the construction of the OptBench evaluation set. It is unclear how the problems were extracted and cleaned from the Optlib library.

The evaluation directly compares a tree-search method against standard reasoning models without normalizing for test-time compute. For proof search, complexity is governed by the number of passes, search width, and depth. Directly comparing this multi-step, verifier-guided search against whole-proof generation models is an inequitable comparison.

This paper diverges from the hyperparameter established by BFS-Prover-V2, which effectively utilizes a narrow expansion width like $k=3$ with higher passes.  Because "stagnant states" essentially act as search-space distractors, the issue of stagnancy might be resolved simply by maintaining a lower expansion width (like changing k=4 to k=3). The paper lacks an ablation study proving that the proposed UAPO objective significantly outperforms a baseline prover running standard BFS with optimally tuned hyperparameters.

---

> ### Author Rebuttal · Authors · 2026-03-30
>
> We thank the reviewer for the suggestions.
>
> - On the construction of OptBench.
>
> All problems in OptBench are extracted from a modified version of Optlib that we adapted for compatibility with the Lean 4.10.0 and LeanDojo-based evaluation environment used by BFS-Prover-V2. We collect theorem and lemma statements from the library, remove purely definitional declarations, bridging lemmas, and entries that are too trivial to be meaningful evaluation targets, and then replay the retained declarations in LeanDojo under the same environment so that every benchmark item can be executed and evaluated consistently. We categorize the resulting benchmark into basics (121), convexity (135), and algorithmic analysis (144) according to the mathematical role of the statement and its source file. The algorithmic subset is also extracted from Optlib. It consists of intermediate proof goals arising in formal convergence analyses of optimization algorithms, rather than problems imported from a separate external source.
>
> - On the comparison between step-level search and whole-proof generation.
>
> Whole-proof generation and step-level proving are different inference regimes, so a strictly fair single-number compute comparsion is difficult. A whole-proof model must commit to a long script in one shot, while a step prover produces short actions and receives verifier feedback after each step. For this reason, in this paper we use step-level Pass@1 and whole-proof Pass@32 only as a practical low-budget reference point: step-level Pass@1 already includes verifier-guided search, whereas whole-proof Pass@1 is much more brittle, so giving the whole-proof model a modest number of samples provides a more reasonable comparison point without moving to heavy test-time scaling.
>
> To make the whole-proof side more explicit, we already report Goedel-Prover-V2 at Pass@32 and Pass@256, and we will additionally extend this evaluation to a larger pass count. The Avg results are summarized as follows:
>
> | Model | Pass@32 | Pass@256 | Pass@512 |
> | --- | --- | --- | --- |
> | Goedel-Prover-V2-8B | 12.50% | 19.75% | 25.75% |
>
> | Model     | Pass@1 | Pass@32 |
> |-----------|--------|---------|
> | OptProver | 37.75% | 55.25%  |
>
>
> These results show that increasing the whole-proof pass budget does help, but the whole-proof regime remains far below the step-level results on OptBench. The main point of this paper is therefore not a perfectly normalized cross-paradigm ranking. Our focus is on bridging Olympiad-style step-level proving and formal optimization, and the main like-for-like comparison for the proposed method is the within-step-level comparison between OptProver and BFS-Prover-V2 under the same search framework.
>
> - On whether the gains may come from search hyperparameters rather than UAPO.
>
> We add a search-width ablation on the EI+PW-UAPO model, comparing $k=3$ and $k=4$ while keeping the remaining search configuration fixed. The table will report both Avg Pass@1 and Avg Pass@32:
>
> | Model | $k$ | Avg Pass@1 | Avg Pass@32 |
> | --- | --- | --- | --- |
> | OptProver (EI + PW-UAPO) | 3 | 36.00% | 54.00% |
> | OptProver (EI + PW-UAPO) | 4 | 37.75% | 55.25% |
>
> If a narrower expansion width were enough to resolve stagnancy, then moving from $k=4$ to $k=3$ should improve performance. We observe the opposite: performance drops from 37.75% to 36.00% at Pass@1 and from 55.25% to 54.00% at Pass@32. This suggests that narrowing the search tree alone does not explain the gain, and the improvement cannot be attributed simply to search-width tuning. All reported step-level results are evaluated under the same best-first search configuration described in the paper: sampling temperature $\tau = 1.3$, and length normalization factor $\alpha = 2.0$, and expansion width $k=4$. This means the gains from EI, DPO, UAPO, and PW-DPO are measured under a fixed search procedure rather than by changing inference hyperparameters across models.

---

> > ### Author Rebuttal · Reviewer_Qe8b · 2026-04-03
> >
> > Thank you for the response. I increased the score.

---

### Official Review · Reviewer_PgnN · 2026-03-13

**Soundness:** 3
**Presentation:** 4
**Significance:** 2
**Originality:** 3
**Overall Recommendation:** 4
**Confidence:** 3

**Summary:**

This paper addresses the problem of migrating formal theorem provers trained on Mathlib (on abstract mathematics) to applied optimization (Optlib), where proof corpus differ hugely. The authors demonstrate that plain fine-tuning on formalized optimization textbooks degrades proving capability, where the model learns syntactically valid but strategically unhelpful tactics that clog proof search. To address this, the paper proposes a three-stage training pipeline:
(1) expert iteration to generate training trajectories in the model's own tactic style,
(2) a utility-aware preference optimization method (UAPO) that penalizes valid-but-stale tactics using a 3-level utility scale derived from search outcomes, and
(3) perplexity-weighted DPO to stabilize training on out-of-distribution tokens.

The authors construct OptBench, a 400-problem benchmark drawn from Optlib covering basic calculus, convexity, and algorithm analysis. OptProver achieves 55% Pass@32 on OptBench (up from 32% for the base model) without degrading performance on general benchmarks (MiniF2F, ProofNet). The ablation shows that expert iteration accounts for most of the improvement, with the preference optimization methods contributing a smaller incremental gain.

**Compliance With Llm Reviewing Policy:**

Affirmed.

**Key Questions For Authors:**

What lemmas and tactics get reused across expert iteration proofs? If a small set of Mathlib lemmas (e.g., gradient rules, norm inequalities) appears in most successful EI trajectories, that would explain why EI works so well on optimization — the domain is pattern-repetitive. Could you provide a frequency analysis of the most commonly used lemmas/tactics in EI-generated proofs?

**Limitations:**

Not adequately discussed. The paper has no explicit limitations section.
Missing:
(1) no significance testing on a 400-problem benchmark where the key method differences are ~5 points;
(2) only one domain transfer tested, "general framework" claim is unsupported
(3) no analysis of whether Olympiad skills actually help optimization or the model just learns Optlib from scratch via EI;

**Strengths And Weaknesses:**

Strengths:
The "interactive distribution shift" finding is clean.
The stagnant tactic idea is convincing: a tactic that compiles but leads nowhere wastes search budget silently, worse than one that just crashes.

Weaknesses:
Expert iteration does most of the work (32% to 50%), which makes sense — optimization proofs tend to reuse the same lemmas and patterns, so once the model finds a few successful proofs the style transfers easily. The preference optimization adds only ~5 more points (50% to 55%) on 400 problems with no confidence intervals, making it hard to justify.
The paper says "bridging Olympiad and optimization" but undergrad optimization is mostly plain calculus + inequalities + KKT; students can memorize the proof logic/pattern and just do it, where the Mathlib-style questions require more expertise or domain knowledge. The real barrier is fighting Lean's type system for unfamiliar Optlib definitions, not mathematical domain transfer. Only one transfer direction tested (Mathlib → Optlib), so "general framework" is oversold.

---

> ### Author Rebuttal · Authors · 2026-03-30
>
> We thank the reviewer for the suggestions.
>
> - On the role of expert iteration and preference optimization.
>
> Expert iteration (EI) is the main source of improvement in our pipeline. At the same time, its gains become saturated after the third round of expert iteration (EI3): adding one more EI round raises OptBench performance to **only 50.5%** Pass@32 under the same hyperparameters, indicating no substantive further improvement. In contrast, preference optimization continues to improve performance beyond this EI-saturated regime, with the best variant, PW-UAPO, reaching **55.25%** Pass@32. We summarize this progression below:
>
> | Stage | EI1 | EI2 | EI3 | EI4 |  EI3 + PW-UAPO |
> | --- | --- | --- | --- | ---|  --- |
> | OptBench Avg Pass@32 | 44.75% | 47.5% | 50% | 50.5% | 55.25% |
>
> This suggests that preference optimization provides a genuine additional benefit, rather than merely replicating the effect of more EI rounds. We therefore view it as an effective second-stage refinement on top of EI, with a consistent additional gain even after EI has largely saturated.
>
> To make the experiments more rigorous, we additionally report a small-scale variance study with 3 random seeds, comparing EI and EI+PW-UAPO on OptBench Avg Pass@32:
>
> | Method | Mean | Std |
> | --- | --- | --- |
> | OptProver (Only EI) | 49.92%  | 2.87 |
> | OptProver (EI + PW-UAPO) | 55.33% | 2.05 |
>
> - On the nature of optimization-domain transfer.
>
> Optimization proofs contain reusable patterns, as does any mathematical domain. OptBench covers more than elementary calculus, inequalities, and KKT-style manipulations. The benchmark covers several layers of undergraduate optimization mathematics: high-dimensional differential reasoning, convex analysis, extended-real arguments, and local steps from algorithmic convergence proofs. This means the model must handle not only standard manipulation, but also the mathematical infrastructure of subgradients, cones, polyhedra, proximal constructions and asymptotic convergence. The transfer is not merely from one proof style to another. It is from Olympiad-style problem solving to a substantially different mathematical domain with its own concepts, abstractions, and proof patterns. In this setting, the challenge is not only Lean engineering or type-system difficulty. It also lies in formal reasoning itself. The model must learn how optimization arguments are organized in a formal proof, and how ideas from calculus, convexity, and algorithms must be connected through precise, verifier-accepted intermediate steps.
>
> - On whether the model is only reusing a small set of fixed patterns.
>
> The tactic distribution is not especially informative here, because the highest-frequency actions are generic proof operators such as `simp`, `exact`, and `intro`. The more relevant signal is theorem/lemma reuse. At that level, the distribution is not concentrated: the top-20 named theorem/lemma references account for only 9.2% of all references, so the EI effect is not well explained by memorizing a tiny set of fixed optimization lemmas.
>
> After excluding generic algebraic and structural facts such as `mul_comm` and `le_trans`, the reused Mathlib lemmas are spread across several optimization-relevant families rather than a tiny core set. Representative examples are summarized below:
>
> | Family | Representative reused lemmas |
> | --- | --- |
> | Analysis | `real_inner_self_eq_norm_sq`, `ContinuousLinearMap.opNorm_le_bound`, `HasGradientAtFilter`, `Metric.tendsto_nhds` |
> | Convexity | `ConvexOn`, `lineMap_apply` |
> | Extended-real reasoning | `EReal.coe_add`, `ENNReal.ofReal_mul` |
>
> This suggests that EI does benefit from reusable local proof patterns, but these patterns are distributed across a broad range of mathematical ingredients rather than a tiny set of repeated optimization facts.
>
> - On the scope of our claim.
>
> Our results show that a step-level prover can be successfully transferred from a broad Olympiad/Mathlib-oriented setting to formal optimization in Optlib. We view this as evidence supporting the framework proposed in the paper. At the same time, since the current paper studies only one transfer direction, we will state the claim more modestly. Our new statement is as follows: The paper provides a practical framework for domain adaptation in step-level theorem proving, together with a representative case study in formal optimization.
>
> - On limitations.
>
> We will add an explicit limitations section focusing on the main scope constraint of the current paper: we evaluate only one transfer direction, from a broad Olympiad/Mathlib-oriented prover to Optlib. We will state this point clearly and frame broader transfer settings as future work.

---

> > ### Author Rebuttal · Reviewer_PgnN · 2026-04-02
> >
> > My question are resolved.

---

### Decision · Program_Chairs · 2026-04-30

**Decision:**

Accept (regular)

**Comment:**

This paper presents OptProver, a continual training pipeline for adapting Olympiad-level step-level theorem provers to formal optimization in Lean 4, along with OptBench, a new 400-problem benchmark. Three reviewers initially rated Weak Accept; the fourth rated Weak Reject. After rebuttal, all four selected "Fully resolved," with two explicitly raising their scores.

The main critique was that expert iteration provides most gains (32% to 50%) while preference optimization adds only ~5pp. The authors convincingly showed that EI saturates after round 3, and PW-UAPO provides meaningful additional gains beyond EI saturation (50% to 55.25%), confirmed by a 3-seed variance study. The identification of the "interactive distribution shift" failure mode is a clear and well-motivated contribution.

With all reviewer concerns resolved post-rebuttal, I recommend acceptance.